# BSBP-RWKV: Background Suppression with Boundary Preservation for Efficient Medical Image Segmentation

Xudong Zhou[*]

Institute of Intelligent Machines, Hefei Institutes of
Physical Science, Chinese Academy of Sciences
Hefei, Anhui
University of Science and Technology of China
Hefei, Anhui
xd_zhou@mail.ustc.edu.cn

Tianxiang Chen[†*]

University of Science and Technology of China
Hefei, Anhui
txchen@mail.ustc.edu.cn

## Abstract

Medical image segmentation is of great significance to disease diagnosis and treatment planning. Despite multiple progresses, most present methods (1) pay insufficient attention to suppressing background noise disturbance that impacts segmentation accuracy and (2) are not efficient enough, especially when the images are of large resolutions. To address the two challenges, we turn to a traditional de-noising method and a new efficient network structure and propose BSBP-RWKV for accurate and efficient medical image segmentation. Specifically, we combine the advantages of Perona-Malik Diffusion (PMD) in noise suppression without losing boundary details and RWKV in its efficient structure, and devise the DWT-PMD RWKV Block across one of our encoder branches to preserve boundary details of lesion areas while suppressing background noise disturbance in an efficient structure. Then we feed the de-noised lesion boundary cues to our proposed Multi-Step Runge-Kutta convolutional Block to supplement the cues with more local details. We also propose a novel loss function for shape refinement that can align the shape of predicted lesion areas with GT masks in both spatial and frequency domains. Experiments on ISIC 2016 and Kvasir-SEG show the superior accuracy and efficiency of our BSBP-RWKV. Specifically, BSBP-RWKV reduces complexity of 5.8 times compared with the SOTA while also cutting down GPU memory usage by over 62.7% for each $1024 \times 1024$ image during inference.

## CCS Concepts

• **Computing methodologies → Image segmentation**.

## Keywords

Medical Image Segmentation, Perona-Malik Diffusion, RWKV

---

[*]Both authors contributed equally to this research.

[†]Tianxiang Chen is the corresponding author and lead the research.

---

**ACM Reference Format:**

Xudong Zhou and Tianxiang Chen. 2024. BSBP-RWKV: Background Suppression with Boundary Preservation for Efficient Medical Image Segmentation. In *Proceedings of the 32nd ACM International Conference on Multimedia (MM '24), October 28-November 1, 2024, Melbourne, VIC, Australia.* ACM, New York, NY, USA, 9 pages. https://doi.org/10.1145/3664647.3681033

## 1 Introduction

Medical image segmentation is a fundamental task in the field of medical imaging analysis, aimed at extracting precise lesion information from various types of medical images. It plays a crucial role in helping healthcare professionals in accurate diagnosis, treatment planning, and monitoring of various diseases and conditions. However, relying on manual diagnosis is often time-consuming and prone to errors. Therefore, there is a need for an automated, efficient and accurate medical image segmentation method to improve the clinical workflow.

In the early years, traditional methods were commonly used, which involved hand-crafted features. However, these methods heavily relied on prior knowledge, were unstable, and inefficient, resulting in limited performance improvement for medical image segmentation. In recent years, deep learning-based methods have significantly improved the performance of medical image segmentation. These methods can be categorized into two main approaches: CNN-based methods [5, 14, 22, 24, 43] and Transformer-based methods [3, 21]. The performance of CNN-based methods in most medical image segmentation tasks is often inferior to Transformer-based methods due to the limitation of CNNs in focusing solely on local features. However, while Transformer-based methods excel in modeling long-range dependencies, they generally suffer from a weak local feature extraction ability [10, 11, 42]. As a result, researchers have turned to hybrid approaches that combine CNNs and Transformers [4, 19, 35, 39]. Using CNNs in extracting local features and the capabilities of Transformers in modeling long-range dependencies, these hybrid methods have achieved state-of-the-art performance in many medical image segmentation tasks.

Although improvements have been made, most of these methods (1) are unaware of the background noise disturbance that sometimes impacts accuracy, and (2) suffer from inefficient segmentation dealing with high-resolution images, which are very common in the medical domain. Actually, the noise disturbance problem is non-negligible in the medical images, as we visualized the PSNR of some randomly sampled medical images in two datasets before and after our de-noising operation in Fig. 1. We also offer a set of visual

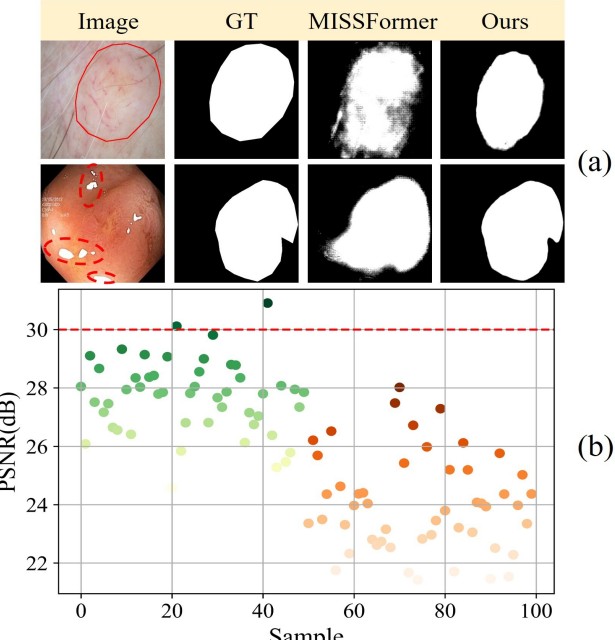

(a)

(b)

**Figure 1: (a) Visualization of the flawed shape-aware segmentation challenge and the boundary disturbance challenge. (b) Computed the peak signal-to-noise ratio (PSNR) between the random sampled 100 original images and the post-PMD denoising images for Kvasir-SEG (green) and ISIC 2016 (orange). The PSNR values above 30dB are considered to indicate less noise interference in the image, while those between 20-30dB are considered to indicate acceptable image quality, but the image is affected by noise.**

comparison results to show that recent well-performed medical image segmentation methods can still be impacted by irrelevant noise and mistake these noise points for lesion areas. In addition, once the image resolution is large, present methods often seem inefficient, as compared in Fig. 2, due to the quadratic complexity of transformer structures. Despite linear complexity designs, their segmentation accuracy are still subordinate to some quadratic complexity transformer-based methods.

To solve the efficiency challenge with the increase in image resolution, we turn to Receptance Weighted Key Value (RWKV) [30] in our network design and are the first to apply Receptance Weighted Key Value (RWKV) [30] structure to the medical image analysis domain. RWKV is a new basic network structure with linear complexity and boasts very low computation complexity. To address the challenge of noise disturbance for improved accuracy, we are inspired by Perona-Malik Diffusion (PMD) [31]. Specifically, we propose a PMD RWKV block based on the discrete wavelet transform (DWT) to suppress background noise disturbance while preserving the boundaries of the lesion target. This block describes the process of evolution of the feature map to the final segmentation mask in medical image segmentation from the perspective of pixel diffusion. However, merely the RWKV design cannot sufficiently

extract enough local features to better preserve the lesion shape features, so we further propose the Multi-Step Runge-Kutta convolutional Block. The Multi-Step Runge-Kutta Block is used due to its high-precision feature extraction capability. It intakes the target body feature as well as the DWT-PMD RWKV Block boundary output and integrates the two to improve the shape-aware segmentation quality. Furthermore, we introduce a shape refinement loss which incorporates frequency information. This loss helps the model escape from local optima and further refines the mask when the predicted mask and ground truth exhibit high similarity in the spatial domain and are difficult to optimize.

Our contributions can be summarized in four folds:

- To the best of our knowledge, we are the first to apply RWKV to the medical image task successfully, providing a new benchmark and valuable insights for future advancements in efficient and accurate RWKV-based methods.
- We devise a DWT-PMD RWKV Block to preserve the boundary details of lesion areas while suppressing background disturbances like noise.
- We propose a Multi-Step Runge-Kutta convolutional Block to provide more local feature components by integrating the preserved boundary cues with the main body features of lesion areas. For further shape refinement, we propose a shape refinement loss function by aligning predicted masks with GTs in both spatial and frequency domains.
- Experiments on ISIC 2016 and Kvasir-SEG prove the superior accuracy and efficiency of our method. Specifically, BSBP-RWKV reduces complexity of 5.8 times compared with the SOTA while also cutting down GPU memory usage by over 62.7% for each $1024 \times 1024$ image during inference.

## 2 Related Work

### 2.1 Medical Image Segmentation

Generally, medical image segmentation methods have two categories: CNN-based methods and Transformer-based methods. In the past few years, CNN-based methods have received widespread attention in the field of medical image segmentation due to CNN's powerful feature extraction capabilities. Among them, UNet [32] has been pioneering and has shown promising results in various medical image segmentation tasks. Subsequently, several variants based on UNet have emerged, such as Attention UNet [28] and UNet++ [44]. These methods propose different improvement techniques on top of the U-shaped structure, enhancing the network's feature capture ability and cross-layer feature fusion capability. In addition, there are also CNN-based methods specifically designed for certain tasks, such as fundus images segmentation [16] and lung infection images segmentation [15]. However, the limited receptive field of CNNs makes it hard to capture long-range dependencies [8]. Consequently, when dealing with larger targets, using CNNs often results in incomplete segmentation.

Recently, transformer-based methods have become popular in medical image segmentation tasks. Due to excels at modeling long-range dependencies, transformer-based methods have outperformed many CNN-based methods. Therefore, they are considered as alternatives to CNNs. ViT [12] is the first work to use a pure transformer for image classification, achieving promising results. Then,

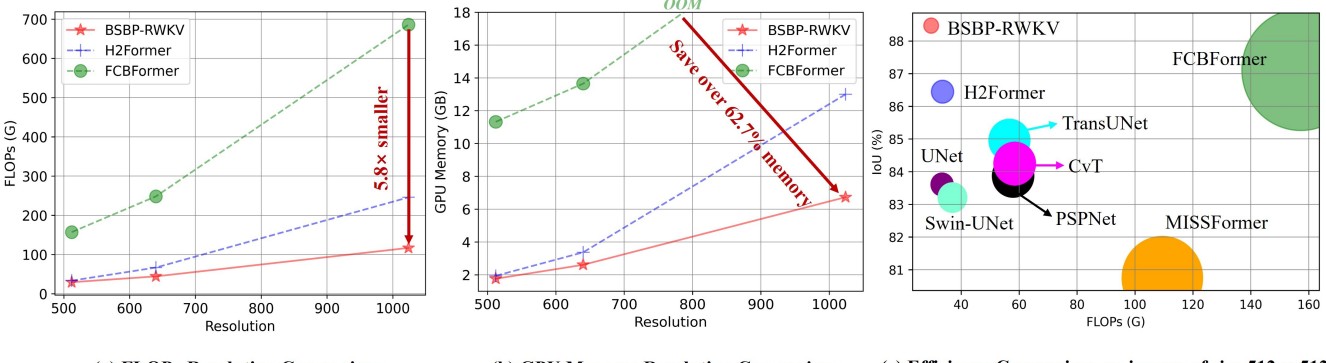

**Figure 2: (a), (b) BSBP-RWKV is more computation and memory efficient than present SOTA methods, H2Former [19] and FCBFormer [34], in dealing with high-resolution medical images. Specifically, When the image resolution is 1024 x 1024, BSBP-RWKV is 5.8× less in complexity and saves over 62.7% GPU memory per image than the SOTA FCBFormer [34]. (c) The overall efficiency comparison on images of resolution 512 × 512, where larger bubbles denote higher computational cost.**

many works based on ViT have been proposed for medical image segmentation. Swin-UNet [3] builds upon the U-shaped structure and reduces computational costs by incorporating local window self-attention, resulting in good performance. TranUNet [4] and H2Former [19] adopt a hybrid CNN-Transformer structure that addresses the problem of pure Transformer structure lacking local spatial information by combining local features from CNNs with global features from Transformers.

However, transformer-based methods have high computational complexity, which limits its application in medical image processing. Receptance Weighted Key Value (RWKV) [30] was initially proposed for natural language processing. It combines the parallel training advantage of transformers with the efficient inference of RNNs, making it a potential alternative to transformers. Vision-RWKV (VRWKV) [13] extends RWKV to computer vision tasks and is the first work to use RWKV for image classification, object detection, and semantic segmentation, showing promising results on these tasks. Therefore, we focus on RWKV-based methods, and to the best of our knowledge, we are the first to introduce RWKV into medical image domain.

The above work either focuses on feature fusion or on constructing effective network modules, while ignoring that some background disturbance may impact segmentation quality and some irregularly shaped lesion targets still cannot get finely segmented. As a solution, in this paper, we describe the process of target pixel refinement in medical image segmentation from the image pixel diffusion perspective and achieve both background suppression and boundary preservation. We also make further attempts for continuous shape refinement through the network by building neural blocks based on the high-accuracy multi-step ODE solver and proposing a novel loss function to align the shape prediction of lesion targets with GT masks in both spatial and frequency domains.

## 2.2 Perona-Malik Diffusion and Neural Network

Perona-Malik Diffusion is an image processing technique based on partial differential equations. This method was originally proposed

by Perona and Malik [31] to address the issues of noise removal and preservation of edge information in images by the introduction of nonlinear anisotropic diffusion. The basic principle of this method is to adjust the diffusion rate based on the gradient differences between pixels in the image, in order to better preserve edge details. The diffusion process of pixels is described by the Perona-Malik equation. In the equation, $(x, y)$ represents the position of the pixel, which gradually diffuses over time $t$ under the guidance of the diffusion coefficient function $g$ (). Ultimately, it reaches an equilibrium state, resulting in the removal of background noise from the image and the preservation of the edges of the targets. The Perona-Malik equation describes how pixels change over time in space. Similarly, in the process of medical image segmentation, as the depth of the network increases, the target pixels gradually transition from coarse to fine under the guidance of the loss function, eventually reaching an equilibrium state.

In recent years, researchers have attempted to establish connections between PMD and neural networks for specific tasks. Ning Wen et al. [38] explored the combination of PMD with CNN for the supervised classification of hyperspectral images. Haowen Zhang et al. [41] integrated PMD into the neural network for low-dose CT denoising. Similarly, Asem Khmag [25] combined the Perona-Malik model with pulse-coupled neural networks for denoising natural digital images. These works explore the potential of combining PMD with a neural network. Ye et al.[40] proposed to combine Mamba with PMD for efficient pediatric left ventricular echocardiographic segmentation. Inspired by the above works, we propose the DWT-PMD RWKV Block, which preserves the target edges while suppressing background noise. This can provide a new perspective on the evolution processes of medical image predicted masks.

## 2.3 Neural Ordinary Differential Equation

Weinan [37] is the first to explore the link between ODE and ResNet [20] and introduced ODE into neural networks. Furthermore, [26, 27] also explained the network from the perspective of ODE. Then, the network inspired by ODE has gained attention and

has achieved remarkable success in image segmentation tasks due to its high-precision feature extraction capabilities. One notable application is its effectiveness in infrared small target detection [6, 7, 9]. However, to our knowledge, ODE has not been widely explored in medical image segmentation tasks. [33] merely combines the classical one-step ODE methods with networks, which inevitably causes some degrees of information loss since only the information from the previous step is used for the prediction of the next step. [29] also attempts to study organ segmentation from the perspective of ODE, but their focus is on the intrinsic robustness of Neural-ODEs rather than accuracy-related issues. Inspired by the above works, we propose a Multi-Step Runge-Kutta Block. Compared with one-step-method-based ODE neural blocks, ours embraces a multi-step memory mechanism, thereby avoiding excessive information loss and offering enhanced feature learning capabilities.

## 3 Method

### 3.1 Overview

We propose BSBP-RWKV, a medical image segmentation framework based on PMD and the Multi-Step Runge-Kutta method. As shown in Fig. 3, the encoder follows a dual-branch structure consisting of a cascaded DWT-PMD RWKV Block feature extraction branch and a parallel Multi-Step Runge-Kutta Block branch. The DWT-PMD RWKV Block feature extraction branch is used to suppress background noise disturbance while preserving the boundaries of the lesion targets, which helps shape-aware segmentation. The Multi-Step Runge-Kutta Block branch aims to integrate the boundary predictions from DWT-PMD RWKV Block branches with target main body features with accurate location cues as a further shape refinement. The shared stem is used to obtain the initial input for the first DWT-PMD RWKV Block and a partial input for the first Multi-Step Runge-Kutta Block. The input to the first Multi-Step Runge-Kutta Block in each stage is composed of the output from the DWT-PMD RWKV Block and the output from the previous stage's Multi-Step Runge-Kutta Block. The resulting feature map is then fed into the plain decoder to progressively enlarge the fused features by four stages until reaching a segmentation head to generate the final mask result. We design a specific loss function to supervise the final prediction.

### 3.2 DWT-PMD RWKV Block

Perona-Malik Diffusion (PMD) is originally used in image de-noising tasks. It can improve image quality by preserving image boundaries and suppressing noise disturbance. Medical images are often corrupted by background noise disturbance and sometimes have blurred lesion area boundaries, which pose great challenges for accurate shape-aware medical segmentation. Therefore, we intend to build a PMD-inspired RWKV block to act upon feature maps so that the background disturbance can be filtered while some target boundary cues can still be preserved.

Given an input feature map $u$, its PMD equation is:

$$\frac{\partial u}{\partial t} = div\left(g\left(|\nabla u|\right) \nabla u\right) \tag{1}$$

where $g|\nabla u|$ is the diffusion coefficient; $t$ is the diffusion step and can be regarded as the layer depth of where the feature map is; $k$

is a positive constant to control the degree of diffusion [17] and is set to 1 by default in our experiments. Notably, Equation (1) is an anisotropic diffusion equation: in the flat or smooth regions where the gradient magnitude is small ($|\nabla u| \to 0$), the diffusion coefficient $g$ is large, meaning that the diffusion is strong and Equation (1) acts as Gaussian smoothing to remove the noise disturbance; in somewhere near the target's boundary, where the gradient magnitude is large ($|\nabla u| \to 1$), the coefficient $g$ is near zero, which means the diffusion is weak so the boundary details can be preserved.

Equation (1) can also be rewritten to the following form:

$$\frac{\partial u}{\partial t} = \frac{\partial}{\partial x}\left\{g\left(\sqrt{\left(\frac{\partial u}{\partial x}\right)^2 + \left(\frac{\partial u}{\partial y}\right)^2}\right)\frac{\partial u}{\partial x}\right\} + \frac{\partial}{\partial y}\left\{g\left(\sqrt{\left(\frac{\partial u}{\partial x}\right)^2 + \left(\frac{\partial u}{\partial y}\right)^2}\right)\frac{\partial u}{\partial y}\right\} \tag{2}$$

where $\frac{\partial u}{\partial x}$ and $\frac{\partial u}{\partial y}$ represent the gradients of the feature map in horizontal and vertical directions.

On the other hand, the Discrete Wavelet Transform (DWT) of an input feature map can can be expressed as:

$$u_i = DWT(u), i \in \{u_{LL}, u_{LH}, u_{HL}, u_{HH}\} \tag{3}$$

where $u_{LL}$ represents the low-frequency component, which primarily reflects the basic structure of the targets in the image. $u_{LH}$, $u_{HL}$ and $u_{HH}$ represent the high-frequency components in horizontal, vertical and diagonal directions of the image, which mainly capture the boundary details. By approximating the derivative terms $\frac{\partial u}{\partial x}$ with $u_{LH}$ and $\frac{\partial u}{\partial y}$ with $u_{HL}$ and setting the diffusion step size $\delta t$ to one, we can convert Equation (2) to the discrete format:

$$u_k = u_{k-1} + \left[g\left(\sqrt{u_{LH}^2 + u_{HL}^2}\right) \cdot u_{LH}\right]_{LH} + \left[g\left(\sqrt{u_{LH}^2 + u_{HL}^2}\right) \cdot u_{HL}\right]_{HL} \tag{4}$$

After enhancing the feature map by PMD, we feed the diffusion output into a RWKV layer implemented by [13]. By piling multiple DWT-PMD RWKV Blocks (shown in Fig. 4) in all layers of one encoder branch, our BSBP-RWKV is equipped with the ability to suppress background noise disturbance while preserving the boundary features of the lesion areas.

### 3.3 Multi-Step Runge-Kutta Block

ODE-based methods have been proven effective in segmentation tasks [6, 33]. However, most methods are based on a single-step ODE solver, which inevitably leads to certain degrees of target feature loss since only the information of one former step is used to make the next step prediction. Inspired by the Multi-Step Runge-Kutta method, we propose the Multi-step Runge-Kutta Block (shown in Fig. 5) that intakes and integrates the boundary output from DWT-PMD RWKV Block and the target body location feature to further refine the shape-aware segmentation quality.

Superior to the Euler method that ResNet is based on and the classic Runge-Kutta method, the Multi-Step Runge-Kutta method is a multi-step ODE solver that achieves third-order prediction accuracy with just two former step predictions. It not only allows for more

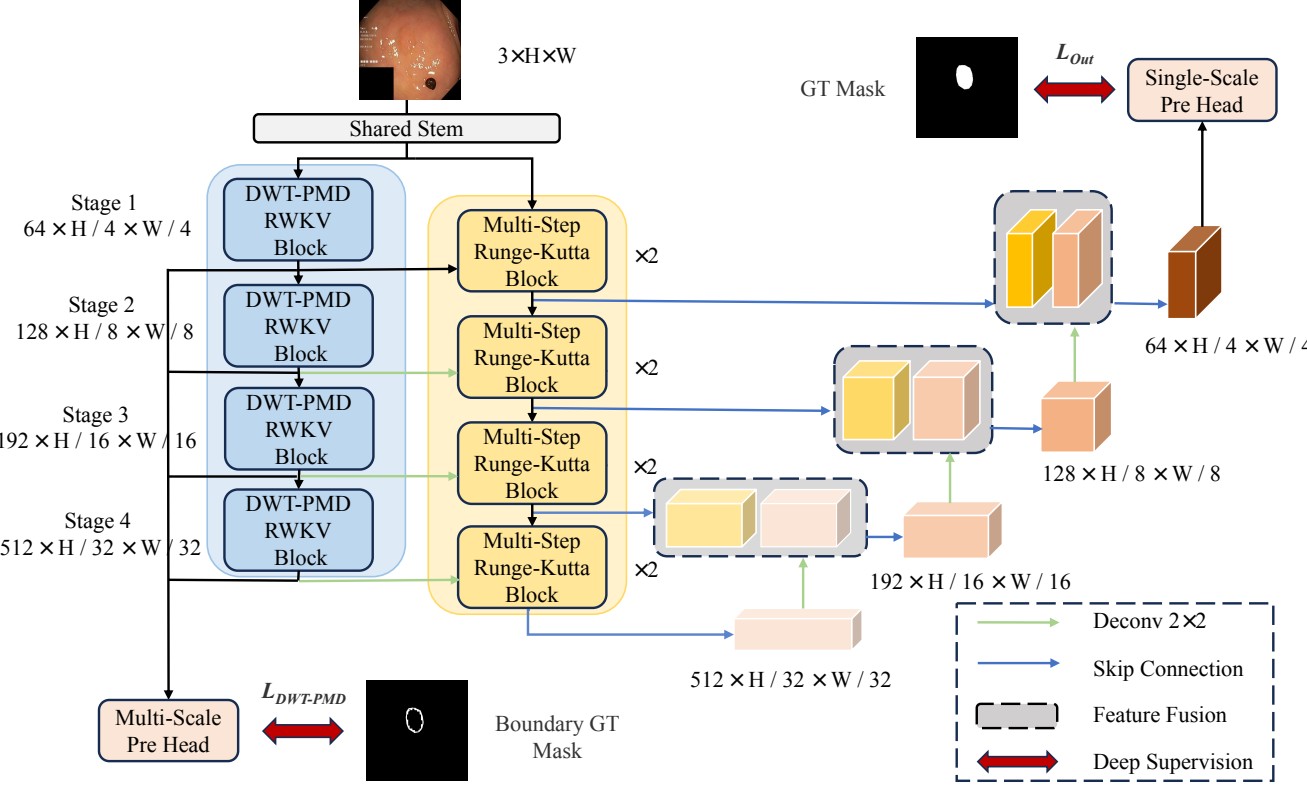

**Figure 3: Overall structure of our proposed BSBP-RWKV, which mainly includes a dual-branch encoder and a plain decoder.**

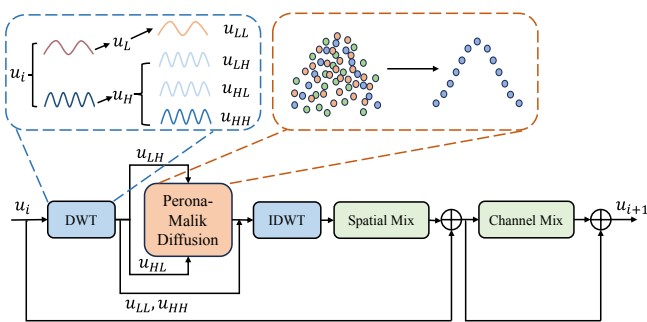

**Figure 4: Structure of our proposed DWT-PMD RWKV Block.**

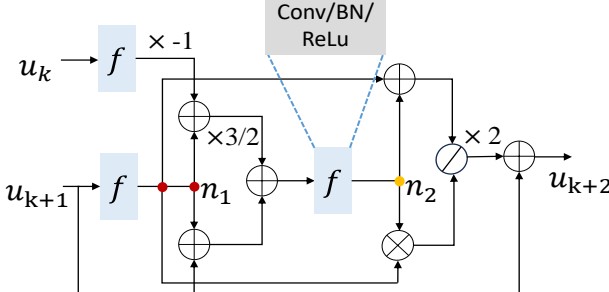

**Figure 5: Structure of our proposed Multi-Step Runge-Kutta Block.**

refined approximations using previous values but has also been demonstrated by [1] to outperform other commonly used multi-step methods like the third-order Adams-Bashforth. The formula of Multi-Step Runge-Kutta method is as follows:

$$u_{k+2} = u_{k+1} + \left(2\frac{n_1 n_2}{n_1 + n_2}\right) \tag{5}$$

where

$$
\begin{aligned}
n_1 &= hf\left(u_{k+1}\right) \\
n_2 &= hf\left(u_{k+1} + n_1 + \frac{3}{2}h\left(f\left(u_{k+1}\right) - f\left(u_k\right)\right)\right)
\end{aligned}
\tag{6}
$$

where $u$ represents the feature map, $k$ denotes the forwarding layer number, and $hf(.)$ represents the basic residual part in ResNet [20]. It is worth noting that we use the boundary output of the current stage's DWT-PMD RWKV Block and the previous layer's Multi-Step Runge-Kutta Block output as $u_k$ and $u_{k+1}$ when predicting $u_{k+2}$. The advanced multi-step memory mechanism of the Multi-Step Runge-Kutta Block enables the network to utilize more previously predicted target shape cues so that the lesion shapes can be progressively refined to better align with the ground truth masks. In this way, the boundary details and the body location feature of the predicted lesion areas get well integrated.

## 3.4 Shape Refinement Loss

Medical image segmentation is a pixel-level classification problem that aims to accurately classify each pixel in an image as either the target or the background. Typically, this problem is addressed using cross-entropy loss and Dice loss, with the latter being proposed by Carole H Sudre et al [36]. for medical image segmentation tasks to address the issue of class imbalance. However, existing loss functions for medical image segmentation are defined in the spatial domain. When the predicted lesion area is iteratively refined to be similar in shape to the GT, it becomes challenging to further optimize the model in the spatial domain, leading to a potential risk of getting stuck in local optima.

To address the above issue, we propose the shape refinement loss by combining a particular frequency loss $\mathcal{L}_{freq}$ with present spatial loss based on DWT, which can be used to highlight the difference between the predicted mask and ground truth using frequency clues to jump out of the local optimum when they become similar in the spatial domain and the model is difficult to further optimize, especially for the boundaries of the lesion area. This is because frequency domain clues are more sensitive to gradient discrepancies at the boundaries of the target and background. Our proposed $\mathcal{L}_{freq}$ is defined as follows:

$$\mathcal{L}_{freq} = \sum_{i \in I} \alpha \left( \phi_1 \left( y_i \right) - \phi_1 \left( \hat{y}_i \right) \right)^2 + \beta \left( \phi_2 \left( y_i \right) - \phi_2 \left( \hat{y}_i \right) \right)^2 \quad (7)$$

where $\phi_1$ and $\phi_2$ denotes the operations of retaining only the low-frequency components and high-frequency components by DWT, respectively. $\alpha$ and $\beta$ are hyper-parameters, and their sum is constrained to be equal to 1. $y$ and $\hat{y}_i$ represent the ground truth and predicted mask, respectively. $i \in I$ refers to a pixel in the $y$ and $\hat{y}_i$.

The spatial loss adopts the combination of cross-entropy loss and Dice loss commonly used in medical image segmentation tasks and are are respectively defined as follows:

$$\mathcal{L}_{space} = (0.3 \mathcal{L}_{ce} + 0.7 \mathcal{L}_{dice}) \quad (8)$$

$$\mathcal{L}_{ce} = -\sum_{i \in I} y_i log \left( \hat{y}_i \right) + (1 - y_i) log \left( 1 - \hat{y}_i \right) \quad (9)$$

$$\mathcal{L}_{dice} = 1 - \frac{2 \sum_{i \in I} y_i \hat{y}_i}{\sum_{i \in I} y_i + \sum_{i \in I} \hat{y}_i} \quad (10)$$

Then we can define our proposed shape refinement loss $\mathcal{L}_{sr}$ as follows:

$$\mathcal{L}_{sr} = \mathcal{L}_{space} + \lambda \mathcal{L}_{freq} \quad (11)$$

where $\lambda$ represents the balance coefficient between the spatial domain loss and the frequency domain loss and is set to 0.8. The overall loss function $\mathcal{L}$ of our BSBP-RWKV includes $\mathcal{L}_{Out}$ as the primary loss and $\mathcal{L}_{DWT-PMD}$ as the auxiliary boundary loss. $\mathcal{L}_{Out}$ follows the $\mathcal{L}_{sr}$ form and $\mathcal{L}_{DWT-PMD}$ follows the $\mathcal{L}_{space}$ form. The overall loss $\mathcal{L}$ is defined as:

$$\mathcal{L} = \mathcal{L}_{Out} + \mathcal{L}_{DWT-PMD} \quad (12)$$

## 4 Experiments

### 4.1 Experimental Settings

*4.1.1 Datasets.* **Skin lesion dataset** [18]. We use the ISIC 2016 skin lesion segmentation dataset, which serves as a benchmark

challenge for automated diagnosis of skin cancer. It contains 900 training images and 379 test images.

**Kvasir-SEG dataset** [23]. This is a largest-scale challenging dataset for gastrointestinal polyp segmentation containing 1000 polyp images. Since the official split setting for training and testing is not provided, we follow the same split setting as in [2], 80% for training and 20% for testing.

*4.1.2 Implementation Details.* All experiments were implemented in the Pytorch and performed on NVIDIA GeForce RTX 3090 GPU. The model was iterated for 250 epochs using the AdamW optimizer with initial learning rate $10^{-4}$, weight decay $10^{-4}$, which is reduced by the "Poly" strategy. The batch size for training is 8. All images and masks are resized to 512 × 512.

*4.1.3 Evaluation Metrics.* For the performance evaluation metrics of the model. We use Dice and IoU, where Dice and IoU represent Dice Similariy Coefficient and Intersection-over-Union, respectively. They are used to measure the similarity between the predicted mask and the ground truth, which can be computed as follows:

$$Dice = \frac{2TP}{(TP + FN) + (TP + FP)} \quad (13)$$

$$IoU = \frac{TP}{FN + FP + TP} \quad (14)$$

where FN, TP, and FP represent false negative, true positive and false positive predictions, respectively.

### 4.2 Comparisons With Other Methods

The quantitative comparison result with other methods is shown in Table 1, where our method achieves the best performances. To be more specific, our BSBP-RWKV achieves a Dice score of 93.51% and an IoU score of 88.47% on the ISIC 2016 dataset, surpassing the second-ranked method FCBFormer by 0.89% and 1.35%, respectively. On the Kvasir-SEG dataset, BSBP-RWKV achieves a Dice score of 92.74% and an IoU score of 87.92%, surpassing the second-ranked method TGDAUNet by 0.67% and 0.93%, respectively. Compared to the Vision-RWKV method, which is also of RWKV type, our approach exceeds it by 3. 28% in terms of Dice score and 4.16% in terms of IoU in the ISIC 2016 dataset. Additionally, on the Kvasir-SEG dataset, our method outperforms Vision-RWKV by 3.3% in Dice score and 4.6% in IoU. The reason why other methods have weaker segmentation performance compared to BSBP-RWKV is that these methods tend to ignore that some image background noise disturbance may disturb the segmentation effect and the importance of target edges. The reason analysis is demonstrated by our visual results shown in Fig. 6 (a), where our BSBP-RWKV is undisturbed by background disturbance and can produce promising segmentation results. Fig. 6 (b) visualizes the effect of our DWT-PMD RWKV Block in boundary preservation and the effect of our Multi-Step Runge-Kutta Block in shape refinement. Equipped with both parts, our loss function brings the best segmentation results.

### 4.3 Ablation Study

The ablation study of each component in BSBP-RWKV is presented in Table 2. From the experiments, we can observe that when the DWT-PMD RWKV Block was used alone, it exhibited an improvement in performance compared to the baseline. This improvement

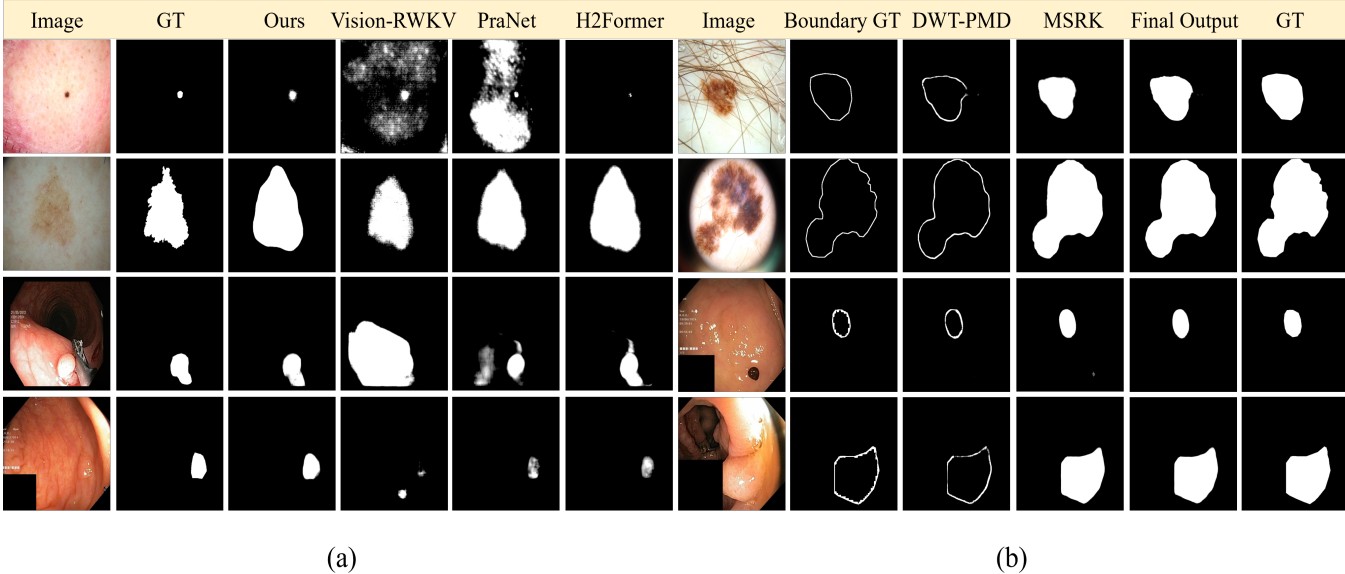

Figure 6: Result visualization of (a) different medical image segmentation methods (b) intermediate branch outputs of BSBP-RWKV.

| Type | Method | ISIC 2016 | | Kvasir-SEG | |
|------|--------|-----------|------|-----------|------|
| | | Dice↑ | IoU↑ | Dice↑ | IoU↑ |
| CNN | UNet | 89.56 | 83.61 | 87.96 | 82.34 |
| | UNet++ | 89.46 | 83.63 | 87.01 | 78.84 |
| | PSPNet | 90.22 | 83.87 | 87.95 | 82.62 |
| | PraNet | 90.20 | 84.73 | 89.80 | 84.00 |
| | nnUNet | 90.45 | 84.52 | 89.75 | 83.59 |
| | DeepLabv3+ | 90.12 | 84.35 | 87.04 | 82.05 |
| | ViG-UNet | 92.06 | 85.58 | 81.88 | 71.04 |
| Trans | Swin-UNet | 90.12 | 83.21 | 70.71 | 60.96 |
| | MISSFormer | 88.37 | 80.75 | 71.56 | 61.17 |
| Hybrid | TransUNet | 91.31 | 84.96 | 89.21 | 83.73 |
| | CvT | 90.42 | 84.25 | 88.13 | 82.04 |
| | TGDAUNet | 92.15 | 86.32 | 92.07 | 86.99 |
| | FCBFormer | 92.62 | 87.12 | 91.43 | 86.03 |
| | H2Former | 92.41 | 86.45 | 91.80 | 86.29 |
| RWKV | Vision-RWKV | 90.23 | 84.31 | 89.44 | 83.32 |
| | BSBP-RWKV | **93.51** | **88.47** | **92.74** | **87.92** |

Table 1: Quantitative results of different methods on ISIC 2016 and Kvasir-SEG in terms of Dice(%), IoU(%).

can be attributed to the DWT-PMD RWKV Block can effectively suppress background noise while preserving boundaries, leading to significant enhancements in the segmentation results. Similarly, when the Multi-Step Runge-Kutta Block was used alone, it also demonstrated an improvement over the baseline. This suggests that the Multi-Step Runge-Kutta Block, with its multi-step memory mechanism, can more accurately extract lesion body features

and thereby improve segmentation results. When the DWT-PMD RWKV Block and Multi-Step Runge-Kutta Block were used together, the segmentation results reached optimum. This suggests that the Multi-Step Runge-Kutta Block can integrate the boundary information learned by the DWT-PMD RWKV Block to achieve better shape-aware segmentation results and also shows the complementary properties of the two modules.

The ablation study on the number of DWT-PMD RWKV Block is shown in Table 3. In this study, we varied the number of blocks in different stages while keeping the total number of blocks in the four stages constant. Experimental results demonstrated that increasing the number of blocks in the intermediate stages, particularly in the third stage, led to improvements in the segmentation performance, rather than evenly distributing them.

Additionally, we ablate the effect of our Multi-Step Runge-Kutta Blcok in Table 4. We compare the performance of Multi-Step Runge-Kutta Blcok with the Single-Step Runge-Kutta Block (SSRK) and the Euler-based Resblock. The performance improvement of the Single-Step Runge-Kutta Block over the Euler-based Resblock is marginal, while the Multi-Step Runge-Kutta Block exhibits a significant performance improvement compared to the Single-Step Runge-Kutta Block. The experiments demonstrate that our Multi-Step Runge-Kutta Blcok with multi-step memory mechanism to integrate target boundary and main body features shows more promising segmentation results.

Fig. 7 ablates the proportion of low-frequency and high-frequency components in the frequency loss ($\mathcal{L}_{freq}$) part of our $\mathcal{L}_{sr}$. The results show that when the low-frequency loss component accounts for around 50% ($\alpha = 0.5$) of the total loss, the segmentation performance is optimal. However, when the low-frequency component is completely absent ($\alpha = 0$) or the high-frequency component is completely missing ($\alpha = 1$), there is a significant decrease in

| Method | ISIC 2016 | | Kvasir-SEG | |
|---|---|---|---|---|
| | Dice↑ | IoU↑ | Dice↑ | IoU↑ |
| Baseline | 90.91 | 85.14 | 89.62 | 83.75 |
| Baseline+MSRK | 92.64 | 87.17 | 90.58 | 85.36 |
| Baseline+DWT-PMD | 92.93 | 87.82 | 91.86 | 86.81 |
| Baseline+DWT-PMD+MSRK | **93.51** | **88.47** | **92.74** | **87.92** |

**Table 2: Ablation study of DWT-PMD RWKV Block and Multi-Step Runge-Kutta Block on ISIC 2016 and Kvasir-SEG in terms of Dice(%), IoU(%). The baseline is RWKV, which follows a U-shaped architecture.**

| DWT-PMD Num | ISIC 2016 | | Kvasir-SEG | |
|---|---|---|---|---|
| | Dice↑ | IoU↑ | Dice↑ | IoU↑ |
| 5, 5, 5, 5 | 92.67 | 87.14 | 91.77 | 86.65 |
| 3, 5, 10, 2 | 93.17 | 87.98 | 92.32 | 87.38 |
| 2, 3, 12, 3 | 93.36 | 88.28 | 92.67 | 87.73 |
| 3, 4, 12, 1 | **93.51** | **88.47** | **92.74** | **87.92** |

**Table 3: Ablation study of DWT-PMD RWKV Block number on ISIC 2016 and Kvasir-SEG in terms of Dice(%), IoU(%).**

| Method | ISIC 2016 | | Kvasir-SEG | |
|---|---|---|---|---|
| | Dice↑ | IoU↑ | Dice↑ | IoU↑ |
| Baseline+ResBlock | 90.84 | 85.17 | 89.78 | 83.77 |
| Baseline+SSRK | 91.25 | 85.29 | 90.13 | 84.14 |
| Baseline+MSRK | **92.64** | **87.17** | **90.58** | **85.36** |

**Table 4: Comparing with other ODE-inspired block on ISIC 2016 and Kvasir-SEG in terms of Dice(%), IoU(%). The baseline is RWKV, which follows a U-shaped architecture.**

performance. This is because low-frequency elements represent the body region of the lesion, while high-frequency elements represent the boundary texture of the lesion area. The complementary nature of the body region and boundary texture is crucial for design of shape refinement loss. We also ablate the shape refinement loss. We also ablate the hyper-parameter $\lambda$ in $\mathcal{L}_{Out}$ on ISIC 2016 in terms of Dice(%) and IoU(%). Using only $\mathcal{L}space$ in $\mathcal{L}Out$ yields a Dice of 93.16% and IoU of 87.88%. Incorporating both $\mathcal{L}space$ and $\mathcal{L}freq$ into $\mathcal{L}_{Out}$ enhances Dice to 93.51% and IoU to 88.47%.

## 4.4 Model Complexity

We also compared BSBP-RWKV with recent methods in terms of computational efficiency and accuracy, as summarized in Table 5. In the last column we present the average IoU (aIoU) results in the ISIC 2016 and Kvasir-SEG datasets. Among these methods, PSPNet has the fewest parameters, while U-Net has the fastest Inference speed. However, their segmentation accuracy performances are mediocre. Similarly, FCBFormer has the highest FLOPs due to its usage of the transformer structure. For Swin-UNet, CvT, and H2Former, although their parameters are similar to ours, their FLOPs are higher

| Method | FLOPs↓ | Params↓ | Inference Time↓ | aIoU↑ |
|---|---|---|---|---|
| UNet | 33.39G | 23.63M | **28.87ms** | 82.98 |
| PSPNet | 57.88G | **14.26M** | 50.39ms | 83.25 |
| Swin-UNet | 36.98G | 27.27M | 34.82ms | 72.09 |
| MISSFormer | 109.45G | 42.33M | 92.86ms | 70.96 |
| TransUNet | 56.66G | 109.54M | 48.65ms | 84.35 |
| CvT | 58.45G | 32.33M | 49.71ms | 83.15 |
| FCBFormer | 157.24G | 52.96M | 118.62ms | 86.58 |
| H2Former | 33.56G | 33.71M | 29.02ms | 86.37 |
| BSBP-RWKV | **29.61G** | 28.09M | 33.24ms | **88.20** |

**Table 5: Comparison study on computational efficiency and accuracy. The image for inference is in size of 512 × 512.**

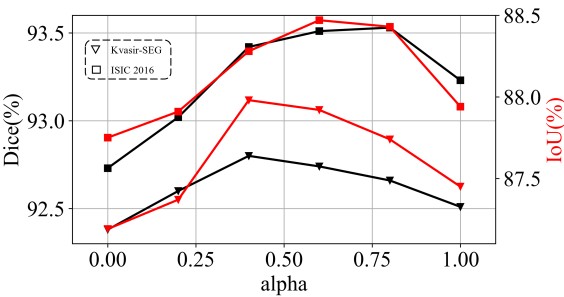

**Figure 7: Effect study of $\alpha$ in $\mathcal{L}_{freq}$ on ISIC 2016 and Kvasir-SEG in terms of Dice(%), IoU(%).**

than ours, and their inference speeds are comparable to or slower than ours. Swin-UNet, CvT, and H2Former have parameters similar to ours, but their FLOPs are higher and their inference speeds are slower than ours. Additionally, their segmentation accuracy is significantly worse than our method. In general, we can observe that our BSBP-RWKV achieves the best trade-off between segmentation fineness and model efficiency. This makes BSBP-RWKV suitable for both accuracy-oriented and efficiency-oriented applications.

## 5 Conclusion

To solve the challenges of background noise disturbance and inefficient segmentation, we turn to RWKV in our network design and propose BSBP-RWKV for efficient medical image segmentation. Specifically, we devise DWT-PMD RWKV Block to eliminate background disturbance while preserving lesion target boundaries. We also propose the Multi-Step Runge-Kutta Block based on an ODE solver to integrate the accurately located target main body feature with DWT-PMD RWKV Block boundary outputs as a further shape refinement manner. In addition, we design a shape refinement loss to evade the network from getting stuck into a local optima in spatial domain during shape refinement. Experimental results on ISIC 2016 and Kvasir-SEG demonstrate the SOTA performance and efficiency of our network. Specifically, BSBP-RWKV reduces the complexity of 5.8 times compared to the SOTA while also reducing GPU memory usage by more than 62. 7% for each 1024 × 1024 image during inference.

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
