# OpenReview forum: "BSBP-RWKV: Background Suppression with Boundary Preservation for Efficient Medical Image Segmentation"
_acmmm.org/ACMMM/2024/Conference — MM2024 Poster_

### Official Review · Reviewer_wBjF · 2024-05-08

**Rating:** 3
**Confidence:** 4

**Summary:**

This paper introduces an innovative medical image segmentation framework named BSBP-RWKV, which aims to address the issues of background noise interference and improve segmentation efficiency by integrating traditional image de-noising methods with an efficient network architecture. The researchers have designed a DWT-PMD RWKV block to preserve the boundary details of the lesion areas and employed a multi-step Runge-Kutta convolutional block to further enhance the segmentation quality. Additionally, a loss function is introduced to refine the shape of the predicted lesion areas in both spatial and frequency domains.

**Strengths:**

1.The proposed BSBP-RWKV network architecture effectively integrates the traditional Perona-Malik diffusion method with the modern RWKV network structure to enhance the accuracy of medical image segmentation.
2.BSBP-RWKV demonstrates lower computational complexity and GPU memory usage when processing high-resolution medical images, making it more efficient for practical applications.

**Limitations:**

1.The model's validation solely on dermatoscopic and endoscopic images renders the term "Medical Image Segmentation" in the title inappropriate, as it implies a broader applicability across multiple modalities. Validation across diverse datasets, including ultrasound, MRI, CT, and pathology images, is imperative for justifying such a claim.
2.The BSBP-RWKV framework, while addressing noise reduction and computational efficiency, lacks significant novelty, primarily combining existing methodologies rather than introducing novel approaches.
3.While Figure 1(b) highlights noise presence in certain PSNR ranges, the paper lacks comprehensive analysis on noise reduction in foreground regions and post-denoising PSNR distribution. Adequate experimental evidence is necessary for thorough validation.
4.The section discussing the loss function lacks the requisite level of detail for a thorough understanding. Specifically, the integration method of multi-scale feature maps remains ambiguous—are they combined through interpolation or deconvolution operations? This clarification is crucial due to the inherent variability in dimensions across feature maps at different stages. Furthermore, the process for selecting weights in the loss function requires elucidation. Notably, existing medical imaging datasets typically lack Boundary Ground Truth (GT) Masks, prompting inquiry into their derivation. Are these masks manually calculated, and if so, does this introduce potential human bias into the evaluation process?
5.The absence of five-fold cross-validation and validation on multi-center datasets raises significant doubts regarding the model's generalizability. Recommending validation on additional datasets like ISIC 2017 and ISIC 2018 or cross-dataset inference can enhance the model's robustness.
6.It is pertinent to inquire whether the reported results on the Kvasir-SEG dataset are derived directly from the competition's leaderboard. A comparison with the leaderboard reveals that the performance of existing methodologies surpasses that of the model presented in this paper. This observation prompts further investigation into the model's efficacy relative to state-of-the-art approaches. Clarification regarding the source of results and a detailed comparison with competing methodologies are essential for an accurate assessment of the model's performance. (https://paperswithcode.com/sota/medical-image-segmentation-on-kvasir-seg)
7.The significant influence of hyperparameter α on model results prompts inquiry into its consistency across different datasets and whether observed performance improvements stem from parameter adjustments rather than inherent model enhancements.

**Suitability:**

2

---

### Official Review · Reviewer_YNqu · 2024-05-21

**Rating:** 4
**Confidence:** 1

**Summary:**

To address the challenge of suppressing the interference of background noise on segmentation accuracy when images have high resolution, the BSBP-RWKV for accurate and efficient medical image segmentation is proposed. Specifically, the advantages of Perona-Malik Diffusion (PMD) in noise suppression are combined with the advantages of RWKV in its effective structure, proposing a multi-level Runge-Kutta convolutional block and a new loss function for shape refinement. Finally, validation was carried out on two datasets.

**Strengths:**

1. This paper claims to be the first method to successfully apply RWKV to medical image tasks, achieving a good balance between computational cost and segmentation performance.

2. The ablation study is quite comprehensive, with an ablation of each module.

**Limitations:**

Figure 6(b) could consider using different colors to further illustrate the performance improvement after adding different modules, as the current effect is not very noticeable at first glance.

**Suitability:**

2

---

### Official Review · Reviewer_T4BQ · 2024-06-04

**Rating:** 4
**Confidence:** 3

**Summary:**

Through the observation that there exists background noise disturbance in medical images and existing Transformer-based models for medical image segmentation are insufficient, the paper designs an efficient network that combine the advantages of Perona-Malik Diffusion (PMD) in noise suppression and Receptance Weighted Key Value (RWKV) in efficient inference. The paper builds RWKV blocks upon Discrete Wavelet Transform (DWT) and PMD for background suppression and boundary detail preservation, and devise a multi-step runge-kutta convolutional block to enhance feature learning. Under the supervision of loss in both spatial and frequency domains, the proposed model surpasses state-of-the-art methods and enjoys lower inference costs on two datasets, ISIC 2026 and Kvasir-SEG.

**Strengths:**

1. The paper has clear motivation. For example, the bottom example of Figure 1(a) clearly show how the background noise disturbance (i.e., the reflection) impacts the segmentation accuracy, whereas Figure 1(b) illustrates the universality of image noises.

2. As shown in Figure 2, the improvements of the proposed model in terms of efficiency and performance are impressive.

3. The paper shows the feasibility of the RWKV-like architecture in medical image segmentation and conducts comprehensive experiments to verify the effectiveness of major components in the proposed model.

**Limitations:**

1\. The full name of the abbreviation “DWT“ in Line 19 should be given.

2\. As stated in line 24, the paper “propose a novel loss function“, whose effectiveness, however, is not fully verified. Although the authors have analyzed the impact of $\alpha$ in Figure 7, the readers do not know the performance improvement brought by the loss applied in the frequency domain (Equation 7).

3\. The role of DWT in the proposed DWT-PMD RWKV Block (Section 3.2) is not clear. Can DWT be removed?

4\. The diagram shown in Figure 5 is confusing. Legends should be given. Besides, symbols in Equation 6 (e.g., $hf$) do not correspond to Figure 5 well.

5\. The descriptions in lines 621-623 and the equation 12 are not clear.

6\. The computational efficiency of Vision-RWKV is not given in Table 5.

7\. The paper lacks some reference papers that utilize discrete wavelet transform for medical image segmentation, e.g., [1], [2], [3].

[1] Zhang, Liping. "A medical image segmentation methods based on SOM and wavelet transforms." *MIPPR 2019: Automatic Target Recognition and Navigation*. Vol. 11429. SPIE, 2020.

[2] Li, Ying, et al. "Wavelet u-net for medical image segmentation." *Artificial Neural Networks and Machine Learning–ICANN 2020: 29th International Conference on Artificial Neural Networks, Bratislava, Slovakia, September 15–18, 2020, Proceedings, Part I 29*. Springer International Publishing, 2020.

[3] Zhao, Yawu, et al. "WRANet: wavelet integrated residual attention U-Net network for medical image segmentation." *Complex & intelligent systems* 9.6 (2023): 6971-6983.

**Suitability:**

3

---

### Official Review · Reviewer_19kn · 2024-06-05

**Rating:** 6
**Confidence:** 3

**Summary:**

The innovative solutions presented in the paper introduce advanced techniques to enhance medical image segmentation. The Receptance Weighted Key Value (RWKV) network structure is applied for the first time in the field of medical image analysis, offering a linear complexity that significantly reduces computational demands. The DWT-PMD RWKV block combines Discrete Wavelet Transform (DWT) with Perona-Malik Diffusion (PMD) to suppress background noise while preserving lesion boundary details effectively. Furthermore, the Multi-Step Runge-Kutta convolutional block leverages its high-precision feature extraction capability to improve the quality of shape-aware segmentation. A novel shape refinement loss function utilizes both spatial and frequency domain information to refine shapes and help the model escape local optima. Experimental validation on ISIC 2016 and Kvasir-SEG datasets demonstrates that this approach surpasses existing state-of-the-art techniques in terms of accuracy and efficiency, significantly reducing computational complexity and GPU memory usage.

**Strengths:**

The article presents a rich and innovative methodology in medical image segmentation, introducing several key advancements. Notably, it pioneers the application of the Receptance Weighted Key Value (RWKV) network structure in the domain of medical image analysis. This approach achieves superior performance over state-of-the-art (SOTA) methods on two datasets, ISIC 2016 and Kvasir-SEG, showcasing its effectiveness. The research is well-supported with comprehensive experimental evaluations, including ablation studies, sensitivity analysis, and particularly, assessments of computational efficiency. These experiments provide a robust validation of the methodology's practical applicability and increase confidence in its real-world usage. Additionally, the article is well-organized with clear and informative charts and diagrams, enhancing its readability and understanding.

**Limitations:**

While the paper reaches a level of innovation and content sufficient for conference acceptance, it still exhibits some flaws that could be addressed to enhance clarity and professionalism. Firstly, although the article is generally well-structured, certain descriptions, such as the explanation of the variable 'k' in section 3.2, lack clarity as the symbol 'k' does not appear in the surrounding context. Secondly, there are minor but notable inconsistencies in the diagrams and formulas; for example, in Figure 3, the 'x2' should be placed inside the MSRK box, not outside, and the notation for the loss function in the figure ('L') does not match the mathematical notation used in the formulas ('mathcal{L}'). Additionally, the absence of punctuation in the formulas makes the paper appear less professional. It is also suggested that references to other methods could be better organized, perhaps in a tabular format. Lastly, the paper describes several tunable hyperparameters without justification for their fixed values, which could complicate the adaptation of the methods to other datasets. Future revisions could consider simplifying these elements to make the methodology more accessible.

**Suitability:**

2

---

### Meta-Review · Area_Chair_xHsa · 2024-07-07

**Recommendation:** Accept (Poster)
**Confidence:** 5

**Metareview:**

Based on the reviewers' agreement following the rebuttal, the meta-review concludes with an Accept decision. This paper introduces the BSBP-RWKV model, which effectively suppresses background noise without compromising boundary preservation in medical image segmentation, demonstrating promising results and is now ready for dissemination within the academic community.